# The Tolerability and Effectiveness of Marine-Based Ingredients in Cosmetics: A Split-Face Clinical Study of a Serum Spray Containing *Fucus vesiculosus* Extract, *Ulva lactuca* Extract, and Ectoin

**Ciska Janssens-Böcker** *, **Karin Wiesweg** and **Claudia Doberenz**

MedSkin Solutions Dr. Suwelack, 48727 Billerbeck, Germany
* Correspondence: ciska.janssens-boecker@medskin-suwelack.com

**Abstract:** Introduction: Marine-derived compounds, such as seaweed extracts, fucoidan and ulvans, and ectoin, have gained attention in recent years due to their unique structural and functional characteristics, which make them attractive ingredients for skincare products. In this study, we developed a serum spray based on fucoidan, *Ulva lactuca* extract, and ectoin and evaluated its efficacy on facial skin. Materials and Methods: A split-face design dermatological evaluation of the serum spray was conducted on 33 subjects with visible signs of skin aging, with 29 subjects completing the study according to its protocol. The subjects had a mean age of 50 years and 16 had sensitive skin. The instrumental efficacy and subjective efficacy of the spray were measured on facial skin by evaluating the trans-epidermal water loss (TEWL), skin pH, skin roughness/wrinkle reduction, and skin hydration at baseline, 20 min after its application and 28 days after its continuous use. Results: We found that the application of the serum spray did not significantly affect the TEWL. The hydration in the area treated with Moisturizer + Spray was 17% higher than that in the area treated with Moisturizer alone after 20 min of application ($p < 0.001$) and 5% higher after 28 days of use ($p < 0.05$). Twenty minutes after the application of the product, the average roughness in the area treated with Moisturizer + Spray decreased significantly, with an average of 7% compared to baseline ($p < 0.001$). With regard to the long-term antiwrinkle effect, 28 days after the continuous use of the product, the average roughness in the area treated with Moisturizer + Spray decreased significantly, with an average of 17% in relation to baseline ($p < 0.001$). The skin pH was significantly lowered by 6% after 28 days of use of the moisturizer + spray ($p < 0.05$). Conclusion: The results of this study suggest that the marine-derived compounds fucoidan, ulvans, and ectoin have hydrating and anti-wrinkle properties that make them effective ingredients for skincare products. The serum spray developed in this study was demonstrated to be safe and increase hydration, showing a reduction in wrinkles and maintenance of the skin barrier function after 28 days of its continuous use. Therefore, it could be a promising addition to skincare products for improving skin health.

**Keywords:** fucoidan; *Ulva lactuca*; ectoin; split face; serum spray

## 1. Introduction

Skincare products have become an essential part of people's daily routines and natural ingredients derived from marine sources have gained increasing attention in recent years. The demand for more natural cosmetics is rapidly increasing, and they are promoted to be green, safer, and more sustainable [1–3]. From the shallows to the depths, the oceans house a vast array of habitats and environmental conditions, abundant in diverse flora and fauna. Various marine systems possess distinctive traits that have spurred biological adaptation, resulting in the creation of a broad range of bioactive molecules. This treasure trove of diversity remains largely unexplored and untapped, forming a living library of untapped potential [1,2].

Seaweeds have long been recognized for their positive impact on skin health. They offer a potential renewable source of bioactive metabolites that possess unique structural and functional characteristics when compared to their terrestrial counterparts. With the growing demand for environmentally conscious natural skincare and cosmetic products, marine-derived compounds, such as seaweed extracts and bioactive compounds such as fucoidan and ectoin, have become increasingly popular in this industry. These bioactive compounds have already undergone some clinical testing and are available on the market [4,5].

Fucoidan is a sulfated polysaccharide found in various species of brown seaweeds, such as *Fucus vesiculosus* and *Undaria pinnatifida*. Fucoidan is extracted from the cell walls of these seaweeds through various methods, such as hot water extraction or enzymatic digestion. Once extracted, it is typically purified and processed into different forms [6,7]. It has gained attention due to its various biological activities, including anti-inflammatory, antioxidant, and immunomodulatory properties [7,8]. Fucoidan has been demonstrated to combat photoaging in vitro in UVB-irradiated HS 68 cells via MMP-1 inhibition and the ERK pathways [4]. Fucoidan is clinically effective for skin soothing and protecting after UV exposure, with a significant reduction in erythema ($-14.7\%$ after 24 h) and TEWL ($-18.3\%$ after 24 h) as compared to placebos [7].

In addition, fucoidan has also been shown to have potential benefits for skin appearance. In a clinical evaluation, 50% of the subjects showed an improvement in their skin brightness, 65% showed a reduction in skin spot appearance, and 45% showed an improvement in the appearance of wrinkles after 60 days of use [7].

*Ulva lactuca* is a green alga that is known by the common name sea lettuce. It lives in harsh environments that are in direct contact with UV light and saltwater and it contains large amounts of flavonoids, tannins, phenols, polysaccharides, and vitamins [9,10]. It has been described as having a protective effect against free radicals, which cause skin aging [11].

Ectoin is a natural amino acid derivate that is synthesized by halophilic bacteria in response to extreme environmental conditions, such as a high salinity, temperature, and UV radiation. Ectoin has been shown to have strong hydrating, anti-wrinkle, and anti-inflammatory properties, making it an attractive ingredient for skincare products [5,12].

Ectoin can protect the skin from environmental stressors, such as UV radiation and air pollution, by stabilizing cell membranes and reducing oxidative stress [13,14].

We developed a serum spray based on a combination of these marine-derived compounds and performed a dermatological evaluation. The objective of this study was to determine its instrumental efficacy, as well as its acceptability and subjective efficacy, on facial skin.

## 2. Materials and Methods

A *Fucus vesiculosus* (fucoidan) extract, *Ulva lactuca* extract, and ectoin were formulated into a serum spray with the following INCI: Aqua, Glycerin, Propanediol, Leuconostoc radish root ferment filtrate, Sea Water, *Fucus vesiculosus* extract, Ectoin, Hydrolyzed *Ulva lactuca* Extract, Sodium Acetylated Hyaluronate, Lactic acid, Dihydroxanthan Gum, and Sodium Citrate (Natural Algae Serum Spray—MedSkin Solutions Dr. Suwelack Ag, Billerbeck, Germany).

The *Fucus vesiculosus* extract (fucoidan) was sourced from Marinova Pty Ltd. in Cambridge, TAS, Australia, the *Ulva lactuca* extract was obtained from Greentech in Saint Beauzire, France, and the ectoin was sourced from Bitop AG in Dortmund, Germany. The concentrations of these ingredients ranged between 0.1 and 5% and were either recommended by the manufacturer or determined to be effective through in vitro or clinical studies.

The test product's safety of use was assessed before the study took place. According to EU cosmetic Regulation no. 1223/2009, a cosmetic product must not cause damage to human health when applied under normal or reasonably foreseeable conditions of use [15].

Therefore, it must be evaluated for its safety of use before human subjects are exposed to it. As a result, further ethical approval was not required.

The study was performed by ZURKO RESEARCH S.L., Spain, in accordance with the ethical principles outlined in the latest version of the Declaration of Helsinki. Thirty-three subjects were enrolled in the study. A hemi-face design was chosen, where the product application area was determined via randomization. It followed a split-face study design: one side was treated with a standardized light moisturizer + Natural Algae Serum spray, and the other side with a standardized light moisturizer only. The serum spray was applied twice a day, with 3 pumps in the hand and then application to the applicable half side of the face before moisturizing.

The total study duration was 28 days. Five days before starting the instrumental measurements, a wash-out phase was conducted using only the light moisturizer on both sides of the face (without the serum spray).

The subjects were included based on the following inclusion criteria: an age range of 30–60 years old, male and female, all skin types (sensitive and non-sensitive), all skin conditions (dry, normal, mixed, and oily), with visible crow's feet and age spots, and in a good health condition. The subjects were excluded if they had dermatological pathologies in the experimental area, cardiovascular, digestive, neurological, psychiatric, genital, urinary, hematological, or endocrine progressive alterations, immunodeficiencies, a previous history of intolerance to medicinal, cosmetic, healthcare, household, or industrial products (especially latex, aluminum, or nickel), a previous history of allergies, photosensitivity or phototoxicity, exposure to intense sunlight or UV during the study, or if they had taken sunbaths or UV rays during the month prior to the study in the test area.

The subjects were instructed not to take sunbaths during the study and refrain from showering 12 h prior to the instrumental measurements. No makeup or other skincare products were allowed during the study (no cleansers or serums, etc.). The face was washed with water only.

*2.1. Cutaneous and Instrumental Measurements*

All the cutaneous measurements were carried out using a hemiface method before the application of the serum spray (D0T0), 20 min after applying the serum spray (DOT20), and 28 days after the continuous use of the serum spray (D28). The measurements included:

- An evaluation of the skin barrier function by assessing the trans-epidermal water loss with the Tewameter TM® 300 (Courage & Khazaka electronic, Köln, Germany),
- An evaluation of the hydration efficacy by measuring the skin capacitance with Corneometer CM® 825 equipment),
- An evaluation of the anti-wrinkle efficacy by measuring the average roughness of a selected wrinkle in the experimental area with PRIMOS-CR®,
- An evaluation of the brightening efficacy by analyzing the content of skin melanin with the Mexameter® MX18 equipment,
- Image support with VISIA®,
- A pH balance assessment using the Skin-pH-Meter® PH 905 (Courage & Khazaka electronic),
- Subject questionnaires to assess the subject satisfaction and subjective effectiveness,
- An assessment of the product tolerability by monitoring any adverse effects.

During the measurements, the panelists who participated in the study stayed in an acclimated room for 20–30 min, with a temperature of $20° \pm 2$ °C and a relative humidity of 40–60%.

Statistical Analysis

Descriptive statistics of the biometric parameters' results are reported for both the treatment and control groups at each experimental time. The reported statistics include the average, standard deviation, and absolute variation, with respect to the moisturizer and baseline. Linear mixed-effects models (LMM) are fitted to evaluate the effect of the

treatment compared to the control at each experimental time. These models consider multiple measures taken over time, which are correlated with each other, by including random effects for each panelist. This approach allows for the intercept to vary randomly between the panelists. In cases where the data cannot be fitted by an LMM, either a Student's t-test is used if the data follow a normal distribution, or the Wilcoxon signed-rank test is used if the data do not follow a normal distribution, to evaluate the effect of the treatment compared to the control at each experimental time. The significance level is established at 0.05 (95% confidence interval) for each statistical test conducted in this study.

## 3. Results

A total of 33 subjects with visible signs of skin aging were included in the study, with 29 subjects (31% male) completing the study according to the protocol. The age of the subjects ranged from 36 to 58 years, with a mean age of 50 years. Among the participants, 16 subjects had sensitive skin (55%).

The barrier function, as measured by the trans-epidermal water loss (TEWL), remained stable and within the normal range. At the 20 min mark after the application of the serum spray, the trans-epidermal water loss in the area treated with Moisturizer + Spray was, on average, 1% higher compared to the area treated with moisturizer alone (not statistically significant). There was also no significant difference compared to the baseline measurement. After 28 days of continuous product use, the trans-epidermal water loss in the area treated with Moisturizer + Spray was, on average, 3% lower compared to the area treated with moisturizer alone (not statistically significant). Furthermore, after 28 days of continuous product use, the trans-epidermal water loss in the area treated with Moisturizer + Spray increased by an average of 7% compared to the baseline measurement (not statistically significant) (Table 1).

**Table 1.** Barrier function—Trans-epidermal water loss (TEWL).

| | Trans-Epidermal Water Loss | | | | | |
| | Moisturizer Only | | | Moisturizer + Spray | | |
| | D0T0 | D0T20 | D28 | D0T0 | D0T20 | D28 |
|---|---|---|---|---|---|---|
| Average (g/h/m$^2$) | 16.08 | 15.58 | 17.22 | 15.58 | 15.78 | 16.63 |
| Standard deviation | 6.99 | 4.48 | 7.33 | 5.78 | 5.60 | 5.66 |
| % of variation to D0T0 | - | −3% | 7% | - | 1% (ns) | 7% (ns) |
| % of variation to Moisturizer only | - | - | - | −3% | 1% (ns) | −3% (ns) |

DOT0 = baseline, DOT20 = after 20 min after first application, D28 = after 28 days, and ns = not significant.

After 20 min of product application, the hydration in the area treated with Moisturizer + Spray showed an average increase of 17% compared to the area treated with only the Moisturizer ($p < 0.001$). Furthermore, after 28 days of continuous product use, the hydration in the area treated with Moisturizer + Spray showed an average increase of 5% compared to the area treated with only the Moisturizer ($p < 0.05$). The hydration of the Moisturizer + Spray, in comparison to the baseline measurement taken 20 min after the product application, showed a significant average increase of 21% ($p < 0.001$). Similarly, after 28 days of continuous product use, the hydration in the area treated with Moisturizer + Spray showed a significant average increase of 9% compared to the baseline measurement ($p < 0.05$) (Table 2).

**Table 2.** Hydration effect (Capacitance).

| | Hydration Kinetics (Capacitance) | | | | | |
| --- | --- | --- | --- | --- | --- | --- |
| | Moisturizer | | | Moisturizer + Spray | | |
| | D0T0 | D0T20 | D28 | D0T0 | D0T20 | D28 |
| Average | 55.67 | 56.93 | 57.31 | 54.99 | 66.48 | 59.97 |
| Standard deviation | 11.77 | 11.25 | 11.75 | 13.84 | 14.68 | 9.71 |
| % of absolute variation with respect to D0T0 | - | 2% | 3% | - | 21% *** | 9% * |
| % of absolute variation with respect to Moisturizer | - | - | - | −1% | 17% *** | 5% * |

* $p < 0.05$; *** $p < 0.001$. DOT0 = baseline, DOT20 = after 20 min after first application, and D28 = after 28 days.

Prior to the product application, the average roughness (PRIMOS—Ra value) in the area where the Moisturizer + Spray was applied showed a significantly higher value, with an average increase of 15% compared to the area where only the Moisturizer was applied.

After 20 min of product application, the average roughness in the area treated with Moisturizer + Spray significantly decreased, with an average of 7% compared to the baseline measurement ($p < 0.001$). The long-term anti-wrinkle effect, observed 28 days after continuous product use, showed a significant decrease in the average roughness of the area treated with Moisturizer + Spray, with an average reduction of 17% compared to the baseline measurement ($p < 0.001$) (Table 3, Figures 1 and 2).

**Table 3.** Skin roughness (PRIMOS—Ra value).

| | Anti-Wrinkles Evaluation | | | | | |
| --- | --- | --- | --- | --- | --- | --- |
| | Moisturizer | | | Moisturizer + Spray | | |
| | D0T0 | D0T20 | D28 | D0T0 | D0T20 | D28 |
| Average | 6.02 | 5.82 | 5.76 | 6.92 | 6.45 | 5.78 |
| Standard deviation | 2.19 | 2.1 | 2.03 | 2.56 | 2.72 | 2.3 |
| % of variation to D0T0 | - | −3% | −4% | - | −7% *** | −17% *** |
| % of variation to Moisturizer | - | - | - | 15% * | 11% | 0.3% |

* $p < 0.05$; *** $p < 0.001$. DOT0 = baseline, DOT20 = after 20 min after first application, and D28 = after 28 days.

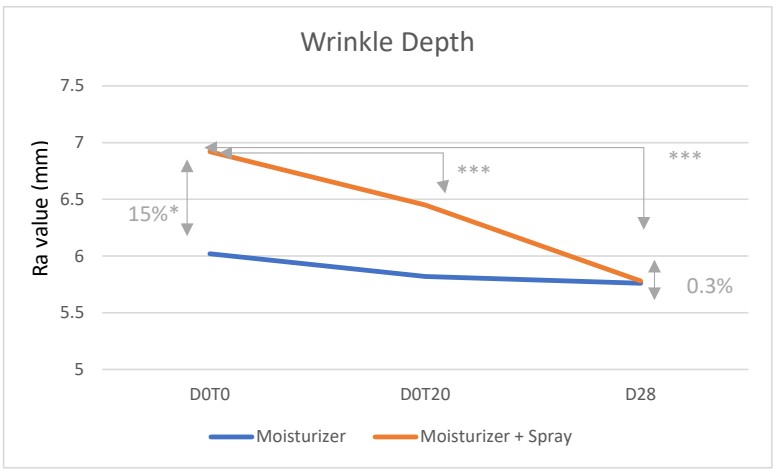

**Figure 1.** Change in wrinkle depth (PRIMOS Ra value, mm). * $p < 0.05$; *** $p < 0.001$.

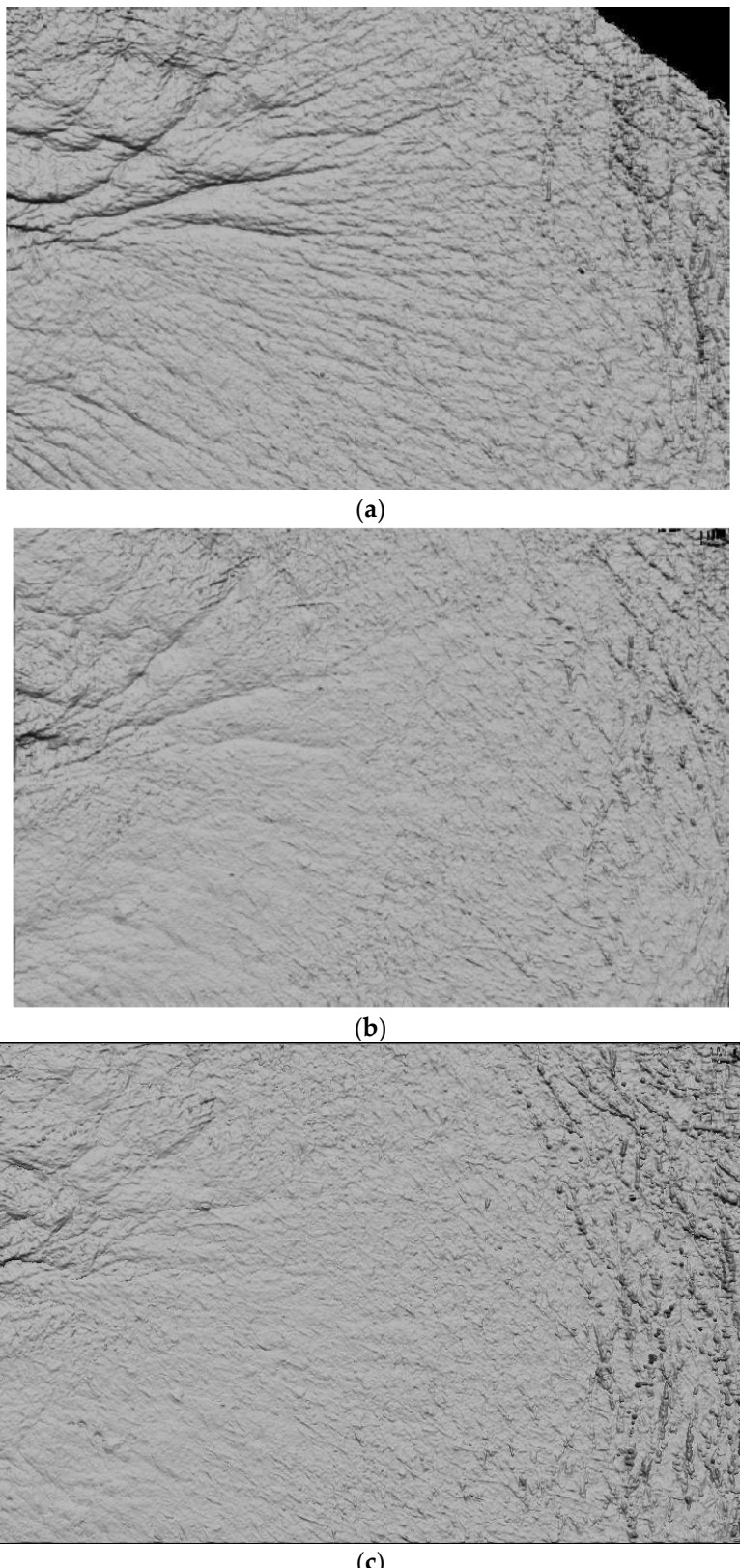

**Figure 2.** Change in wrinkle depth at baseline (**a**), after 20 min (**b**), and after 28 days (**c**) of a representative subject—orbital area (PRIMOS).

After 28 days of continuous product use, the melanin index in the area treated with Moisturizer + Spray showed an average increase of 0.2% compared to the area treated

with only the Moisturizer (not significant). Additionally, on the side treated with Moisturizer + Spray, the melanin index increased by an average of 4% compared to the baseline measurement, although this difference was not statistically significant.

The skin pH in the area treated with Moisturizer + Spray showed a significant average decrease of 6% compared to the baseline measurement after 28 days of continuous product use ($p < 0.05$) (Figure 3). The average skin pH level reached a value of pH < 5.

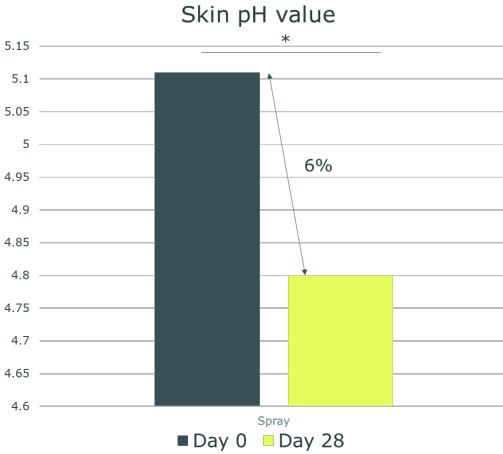

**Figure 3.** Change in skin pH value Moisturizer + Spray. * $p < 0.05$.

According to subjective questionnaires completed after 28 days of using the Moisturizer + Spray, 86% of the subjects noticed improved skin luminosity, 86% felt that their skin was hydrated, 69% observed a reduction in spots, 69% reported smoother wrinkles, and 72% of the subjects felt that their skin appeared plumper. No adverse effects or reactions were reported or observed during the study duration.

## 4. Discussion

Skin is exposed to various external agents that contribute to skin aging, with oxidative stress playing a key role in this. Reactive oxygen species (ROS) are the primary culprits of oxidative stress, causing cellular damage and impairing the skin's appearance [16,17]. This oxidative degradation of lipid membranes negatively affects the skin's overall condition and, as the body ages, its ability to regulate these ROS becomes compromised, leading to an increase in mitochondrial production and, ultimately, skin aging. To counteract this process, the incorporation of antioxidants into cosmetic formulations becomes crucial to mitigating oxidative stress and its detrimental effects [18]. Previous studies have highlighted the significance of antioxidants in combating skin aging.

Exposure to solar radiation is a major source of ROS generation and UV rays can hinder the activity of antioxidant enzymes, further exacerbating oxidative stress [19,20]. Consequently, prolonged exposure to solar radiation is considered to be the primary contributor to skin aging, resulting in manifestations such as hyperpigmentation and photoaging, characterized by a degradation of collagen and hyaluronic acid, ultimately leading to the formation of wrinkles and dry skin [1,21].

In this context, the use of marine-derived ingredients holds promise due to their rich antioxidant properties and diverse bioactivities, which can counteract the damaging effects of reactive oxygen species (ROS) [20]. Macroalgae, commonly known as seaweeds, encompass various taxa such as the *Rhodophyceae* (red algae), *Phaeophyceae* (brown algae), and *Chlorophyceae* (green algae) phyla [22–24]. These organisms have been extensively studied due to their biodegradable and non-toxic natures, making them a valuable source of natural compounds with diverse bioactivities [5,20]. Seaweeds, rich in polysaccharides such as fucoidan, possess antioxidant, immunomodulatory, and skin-aging-delaying properties [24–28].

Our study specifically focused on the efficacy of a serum spray containing *Fucus vesiculosus* extract, *Ulva lactuca* extract, and ectoin, all originally derived from marine sources. These ingredients have been extensively studied and demonstrated potential benefits for skin health, including delaying skin aging, protecting against oxidative stress, and maintaining skin barrier function [7–14]. Previous research has highlighted the bioactivities of seaweeds, emphasizing their ability to delay skin aging, exhibit antioxidant activities, and possess immunomodulatory properties [5,24]. Seaweeds are also known for their high polysaccharide content, such as fucoidan, which possesses antioxidant and anti-inflammatory properties and can be incorporated into various cosmetic products [7,8].

By incorporating these marine-derived ingredients into our formulation, we aimed to harness their antioxidant properties and evaluate their impact on various skin parameters, including hydration, barrier function, and wrinkle reduction. The combination of *Fucus vesiculosus* extract, *Ulva lactuca* extract, and ectoin in our serum spray offered a unique blend of bioactive compounds with potential synergistic effects. This allowed for a comprehensive approach to addressing multiple aspects of skin aging and combating the damaging effects of ROS.

Studies focusing on fucoidans, derived from seaweed extracts, have demonstrated their potential as cosmetic ingredients with anti-aging activities. In vitro studies have revealed that fucoidan can inhibit collagenase and elastase enzymes, which play key roles in skin aging [28]. Moreover, fucoidan has shown immune-regulating properties and been found to increase the levels of the SIRT1 protein, which is involved in maintaining skin function [7]. The polyphenol-rich extract from *Fucus vesiculosus* has demonstrated a significant total antioxidant value, offering protection against oxidative damage on the skin surface [7,29]. Additionally, fucoidan has been shown to promote fibroblast proliferation, collagen deposition, and protect the elastic fiber network of the skin in vitro [28,30]. It has also exhibited potential in regulating the activity and secretion of matrix metalloproteinases (MMPs), inhibiting tyrosinase activity, and displaying anti-melanogenesis properties [28,30,31]. These anti-aging properties align with the findings from our study, in which we found that the serum spray contributed to improvements in hydration and wrinkle reduction.

Pre-clinical and clinical applications have further supported the efficacy of macroalgae-derived ingredients in cosmeceuticals. For instance, *Fucus vesiculosus* has demonstrated soothing and protective effects against UV damage in vivo [7]. A systematic review of the scientific literature also highlighted the clinical safety and efficacy of macroalgae-based skincare products, particularly in terms of their moisturizing, anti-melanogenic, and anti-cellulite benefits [8]. Similarly, *Ulva lactuca*, a green alga, has shown significant antioxidant activity and the ability to protect skin cells from oxidative damage caused by UV radiation [32]. It has also exhibited anti-inflammatory properties and the capacity to stimulate collagen production, thus potentially reducing the appearance of fine lines and wrinkles [33–35].

In our study, we specifically evaluated the efficacy of the serum spray formulation compared to moisturizer alone, focusing on skin barrier function, hydration, and skin roughness. Our findings demonstrated that the Moisturizer + Spray formulation exhibited a significant anti-wrinkle effect, leading to a reduced skin roughness in comparison to the baseline measurements. The barrier function of the subjects remained stable and within the normal range, as indicated by the trans-epidermal water loss measurements [36]

In addition, the serum spray effectively lowered the skin pH to an average pH of <5. pH is a crucial factor in various metabolic, molecular, and cell-regulating processes, particularly in the stratum corneum, where it is essential for the skin's physical, chemical, and microbiological barrier function [37–41]. The skin's pH varies across different regions of the body, typically ranging from moderately acidic (pH 4.1–5.8) as the "normal" pH of the skin surface, with a mean of pH 4.9 [42]. The skin's pH is influenced by factors such as age, gender, race, anatomical position, circadian rhythm, sebum production, skin moisture, and sweating [43]. Various external factors, including occlusion, exposure to skin irritants

such as soaps and detergents, and skin washing, affect the skin's pH. Additionally, the application of cosmetic products can elevate the skin's pH, as many of them have a higher pH than the skin's natural pH [44]. Even the use of tap water, which often has a pH of around 8.0 in Europe, temporarily increases the skin's pH for up to 6 h before returning to its natural level below 5.0, on average [45].

The serum spray was formulated with a low pH and resulted in a lowering of the skin's pH by 6% after 28 days. This was in line with the results that we previously reported, in that buffered skin care products formulated to a pH of ≤ 4.5 can acidify and maintain the physiological skin pH [46]. Daily usage of the serum spray could therefore result in improved skin barrier function, as long-term treatment with skin care products adjusted to a pH of 4.0 has been found to significantly improve the epidermal barrier function compared to identical products with a pH of 6.0 [47].

Hydration plays a crucial role in enhancing the firmness and elasticity of skin [48]. When skin lacks proper hydration, it can experience accelerated desquamation. Moisturizing is a critical part of skincare and has a positive effect on enhancing skin barrier function, metabolism, and appearance. From an aesthetic point of view, dryness of the skin can lead to some undesirable experiences that can undermine a person's confidence, such as pain, itching, tingling, stinging, and uncomfortable sensory feelings or redness, dry white patches, crackers, and even a fissure appearance, or uneven and rough tactile feelings [21,48]. The serum spray in this study contributed to improving the hydration levels in comparison to baseline, with a significant moisturizing effect being observed.

Fitton et al., in their clinical study on 20 subjects, found that *Fucus vesiculosus* extract reduced the melanin index of age spots and increased skin brightness [7]. In contrast, we did not observe a significant skin-lightening effect, which could be attributed to the study period coinciding with increased sunlight exposure.

It is important to consider the potential contributions of the other components present in the product on the observed results in this study, such as Sodium Acetylated Hyaluronate and Lactic acid. However, the concentrations of these ingredients in the serum spray were at low levels, outside of clinically effective ranges, and were included in the formulation for their humectant or buffering properties. Therefore, it is unlikely that the Sodium Acetylated Hyaluronate and Lactic acid significantly influenced the observed properties in our study. The prominent results observed can be attributed primarily to the presence and efficacy of the marine-derived ingredients, namely *Fucus vesiculosus* (fucoidan) extract, *Ulva lactuca* extract, and ectoin. The active marine compounds were incorporated into our formulation at concentrations that have been previously reported to demonstrate bioactive effects [4–13].

While our study utilized a split-face design and standardized moisturizer to mimic a daily skincare routine, the lack of randomization and placebo-controlled intervention may have introduced bias, particularly in the subjective questionnaire assessment. Additionally, the study duration of 28 days may have been relatively short for capturing substantial changes in the skin.

Our study demonstrated the hydrating and anti-wrinkle properties of a serum spray containing *Fucus vesiculosus* extract, *Ulva lactuca* extract, and ectoin, when used in combination with a moisturizer. This study is particularly relevant, as it is the first to investigate the synergistic effects of these three marine-derived ingredients. While previous research has mainly focused on individual marine ingredients, there are limited clinical data available for the combination of these ingredients. Our findings contribute to the growing body of knowledge on marine-derived skincare formulations and highlight the potential benefits of utilizing multiple marine ingredients together. Further research is warranted to explore the long-term effects and mechanism of action of this unique combination in skincare applications.

## 5. Conclusions

In conclusion, the skin is constantly exposed to various external factors that contribute to skin aging, with oxidative stress, which is caused by reactive oxygen species (ROS), being a major culprit.

Natural marine-derived compounds, such as seaweed extracts, fucoidan, ulvan, and ectoin, have emerged as promising active ingredients in skincare products due to their unique properties and potential benefits for skin health. Their utilization in cosmetic formulations has gained significant attention, driven by consumer demand for environmentally friendly and natural products. In our study, the formulation of a serum spray incorporating fucoidan, ulvan, and ectoin yielded positive outcomes in terms of maintaining a healthy skin barrier function, improving hydration, and reducing wrinkles. These marine-derived compounds demonstrated the potential to serve as key ingredients in the development of cosmeceuticals and nutricosmetics.

Nevertheless, further research and rigorous clinical trials are required to fully explore and unlock the complete potential of these natural, marine-derived compounds in the cosmetic industry. By delving deeper into their mechanisms of action and conducting comprehensive investigations, we can enhance our understanding of these compounds and their applications, ultimately advancing the development of effective and sustainable skincare products.

**Author Contributions:** Methodology, C.J.-B.; supervision, C.D.; writing—original draft, C.J.-B.; writing—review and editing, K.W. and C.D. All authors have read and agreed to the published version of the manuscript.

**Funding:** This research received no external funding.

**Institutional Review Board Statement:** Not applicable.

**Informed Consent Statement:** Informed consent was obtained from all subjects involved in the study.

**Data Availability Statement:** The data presented in this study are available on request from the corresponding author. The data are not publicly available due to confidentiality.

**Conflicts of Interest:** The authors are affiliated with MedSkin Solutions Dr. Suwelack, the manufacturer of the product detailed in this manuscript.

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
