# Peer review of "The Tolerability and Effectiveness of Marine-Based Ingredients in Cosmetics: A Split-Face Clinical Study of a Serum Spray Containing Fucus vesiculosus Extract, Ulva lactuca Extract, and Ectoin"

_cosmetics, doi:10.3390/cosmetics10030093_

Round 1
Reviewer 1 Report
The manuscript presents an interesting study on the properties of a spray serum formualted with fucoidan, ulvan and ectoin. The topic is interesting and the results can offer valuable information to the reader, but some corrections are needed to improve the manuscript.
The formula of the product is not presented, the authors should justify why those proportions have been selected. Similarly, the source of the ingredients should be mentioned.
Could other components of the product (Sodium Acetylated Hyaluronate, Lactic acid) contribute to the observed properties?
The discussion section should be rewriten to explain the results obtained in relation to those of other studies. In its present forms it resembles more an Introduction than a Discussion.
Spelling: Latin names in italics, oxadative stress, vesiculosous,
Rewrite lines 252-253, 270-271
Some sentences need rewriting
Author Response
- The formula of the product is not presented, the authors should justify why those proportions have been selected. Similarly, the source of the ingredients should be mentioned.
Response to point 1: A paragraph is added to the methods sections that describes: “The Fucus vesiculosus extract (fucoidan) was sourced from Marinova Pty Ltd in Australia, the Ulva lactuca extract was obtained from Greentech in France, and the ectoin was sourced from Bitop AG in Germany. The concentrations of these ingredients ranged between 0.1-5% and were either recommended by the manufacturer or determined to be effective through in-vitro or clinical studies.”
Due to the fact that the formulation is planned to be launched as a commercial product, the exact concentrations are confidential
- Could other components of the product (Sodium Acetylated Hyaluronate, Lactic acid) contribute to the observed properties?
Response to point 2: The concentrations of these ingredients in the serum spray were at low levels outside of clinically effective ranges and were included in the formulation for their humectant or buffering properties. Therefore, it is unlikely that Sodium Acetylated Hyaluronate and Lactic acid significantly influenced the observed properties in our study. The prominent results observed can be attributed primarily to the presence and efficacy of the marine-derived ingredients, namely Fucus vesiculosus (fucoidan) extract, Ulva lactuca extract, and ectoin. The active marine compounds were incorporated into our formulation at concentrations which have been previously reported to demonstrate bioactive effects.
This is now also described in the Discussion section.
- The discussion section should be rewriten to explain the results obtained in relation to those of other studies. In its present forms it resembles more an Introduction than a Discussion.
Response to point 3: The discussion section is rewritten.
- Spelling: Latin names in italics, oxadative stress, vesiculosous,
- Rewrite lines 252-253, 270-271
Response to point 4&5: manuscript has been rewritten and corrected where appropriate.
Reviewer 2 Report
- add study design to the title
- Latin names in italic
- Proofread the text, English language, double spaces, different fonts
- Please share p values in the abstract for relevant results
- Line 60 - explain if this was a clinical trial and show exact results
- Line 63 - it is has been. Please improve English language
- Line 152 - never start a sentence with a number
- Table 1 - decimal numbers in English language are written with full stop, also is this a different font?
Poor!
Author Response
Response to reviewer 2
Response to assessment of the items in the table that were judged to “Must be improved”:
The introduction was partially rewritten, and more details were provided with references. A paragraph including references was taken out to reflect better the relevant content.
The description of the research design and methods was rewritten to explain the design and set up better. The results section has been revised and p values were added to the text. The discussion and conclusion section was revised and changed.
Comments and Suggestions for Authors
- add study design to the title
Response to point 1: Title is changed to include the clinical design. The split face design was already part of the title : THE TOLERABILITY AND EFFECTIVENESS OF MARINE-BASED INGREDIENTS IN COSMETICS: A SPLIT-FACE CLINICAL STUDY OF A SERUM SPRAY CONTAINING FUCUS VESICULOSUS EXTRACT, ULVA LACTUCA EXTRACT AND ECTOIN
2. Latin names in italic
Response to point 2: This is corrected throughout the manuscript
3. Proofread the text, English language, double spaces, different fonts
Response to point 3: Manuscript is proofread and sections are rewritten and corrected where apppropiate
4. Please share p values in the abstract for relevant results
Response to point 4: p-values were added to the abstract
5. Line 60 - explain if this was a clinical trial and show exact results
Response to point 5: Introduction section has been rewritten at certain parts and more details are provided.
6. Line 63 - it is has been. Please improve English language
Response to point 6: English language was corrected
7. Line 152 - never start a sentence with a number
Response to point 7: Sentence was rewritten
8. Table 1 - decimal numbers in English language are written with full stop, also is this a different font?
Response to point 8: Tables are corrected and font adjusted
Round 2
Reviewer 2 Report
thank you for your improvements